# Vaginal birth core information set: study protocol for a Delphi study to achieve a consensus on a 'core information set' for vaginal birth

Andrew Demetri [1], Anna Davies [1], Danya Bakhbakhi [1], Sharea Ijaz [1], Sarah Dawson,[1] Sheelagh McGuinness,[1] Gemma Beasor,[2] Gemma Clayton,[1] Abigail Johnson,[3] Chloë de Souza,[3] Aine Dempsey,[4] Gabriella Snook,[3] Andrew Sharp,[5,6] David Lissauer,[5] Emma McGoldrick,[6] Christy Burden,[1] Abi Merriel [1,5,6]

¹University of Bristol, Bristol, UK
²Patient Representative, Bristol, UK
³North Bristol NHS Trust, Bristol, UK
⁴University Hospitals Bristol and Weston NHS Foundation Trust, Bristol, UK
⁵Centre for Women's Health Research, Department of Women's and Children's Health, Institute of Life Course and Medical Sciences, University of Liverpool, Liverpool, UK
⁶Liverpool Women's NHS Foundation Trust, Liverpool, UK

**Correspondence to**
Dr Abi Merriel;
abi.merriel@liverpool.ac.uk

## ABSTRACT

**Introduction** Studies have shown that women are often underinformed about potential benefits and risks of vaginal birth. This is in contrast to other modes of birth, such as caesarean birth, for which the risks/benefits are often conveyed prior to undergoing the procedure. A core information set (CIS) is an agreed set of information points that should be discussed with all patients prior to undergoing a procedure or intervention. This CIS could improve the quality of information given regarding mode of birth options, as women will be given information prioritised by patients and stakeholders regarding vaginal birth, empowering them to make informed decisions about their birth. We aim to describe the protocol for the development of this vaginal birth CIS.

**Methods and analysis** We will develop the CIS by: (1) Compiling a 'long-list' of information points about vaginal birth by: undertaking a scoping review of studies and patient information leaflets; interviews with antenatal/postnatal women, an online survey of stakeholders. (2) Collating the 'long-list' of information points and developing the Delphi survey. Think-aloud interviews will refine the survey. (3) Conducting a two-round Delphi survey. 200 stakeholder participants will be recruited. Items rated critically important by ≥80% of participants in one stakeholder group, or with no consensus, will be carried through to a stakeholder consensus meeting to decide the final CIS. Planned start date is 1 June 2022. Planned end date is 31 August 2023.

**Ethics and dissemination** This project has been given a favourable ethics opinion by the University of Bristol Research Ethics Committee (Ref: 10530). Approval from the ethics committee will be sought for any protocol amendments, and the principal investigator will be responsible for these changes. Findings will be presented at relevant conferences and published in a high-impact journal. We will disseminate the CIS, via Policy Bristol, to clinical policy and guideline developers.

## STRENGTHS AND LIMITATIONS OF THIS STUDY

⇒ Patient involvement is central to the development of this body of work, meaning that the core information set will reflect the beliefs and thoughts of the women it will be helping.
⇒ These methods, including the use of in-depth qualitative interviews, the Delphi survey and consensus process, means the development of this core information set will be robust and comprehensive, adding weight to the findings.
⇒ The qualitative interviews, Delphi survey and consensus meetings will only be conducted in the English language due to funding and time limitations.
⇒ The qualitative interviews to help with the formulation of the long list will only be with antenatal and postnatal women, and not the other stakeholder groups, however, their views will be incorporated using the online survey to ensure their views are not missing from the list.
⇒ Due to time and financial constraints, in-depth qualitative interviews will only include women who have given birth in the last 5 years, so may miss some longer-term information points relating to vaginal birth. However this will be mitigated by the scoping review.

## BACKGROUND

Spontaneous vaginal birth (SVB) accounts for approximately 50% of births in the UK.[1] Its benefits, risks and range of experiences are often not discussed in detail with women prior to them opting for a vaginal birth (VB). This is perhaps because VB is the presumed choice for women, unless they have an indication for, or request, a caesarean birth (CB). Recent work has looked at improving informed choice regarding birth method, as well as developing decision aids for women choosing between vaginal or CB.[2–4] To

improve the quality of maternity care, further work must be done to ensure women are provided with the information they require to make a fully informed choice about their planned mode of birth.

VB is a physiological process and has benefits for mother and baby. Examples include shorter hospital stays, quicker recovery times and increased rates of breast feeding compared with CB.[5 6] Risks of VB include shoulder dystocia (baby's shoulders getting stuck once its head has been born) which may lead to birth trauma or asphyxia, and third or fourth degree tears which can result in longer term pelvic floor dysfunction.[7] These outcomes may have significant impact on women's long term physical and psychological functioning.[8]

However, even if a woman has made a positive informed choice for a VB, it may not be successful. In 2020–2021, National Health Service (NHS) statistics showed that 15% of women required an emergency instrumental delivery, and 19% an emergency CB, and most of these will have been aiming for a VB.[9] The risks of emergency CB are increased compared with elective CB, including having a higher incidence of wound infection and postpartum haemorrhage.[7 10] Likewise, instrumental births carry higher risks than SVB, including increased risk of perineal trauma.[7 10] Knowledge about VB, including risks and benefits, may influence women's choices on their mode of birth and birth experience. However, women are generally not explicitly offered information or this choice.

There is a lack of, and inconsistency in, information that women receive about VB.[5] This lack of information antenatally can contribute to poorer birth experiences by increasing anxieties with 'fear of the unknown'.[11 12] The landmark Montgomery case,[13] good clinical practice and NHS improvement support the provision of information for patients to enable them to make truly informed decisions.[14 15] By not providing this, healthcare professionals are failing in the professional, moral and legal requirement, to support decision-making about birth.

Women need access to consistent, accurate information, about what is important to them, when making birth choices. While work has been done to improve information provision for women about CB,[16] very little has been done to understand what information women need about VB to make decisions about mode of birth. Core information sets (CISs) have been developed as a systematic means to provide information that is judged to be important by all relevant stakeholders about medical procedures. A CIS[17 18] is a list of key information points regarding an experience that should be discussed with a person prior to experiencing an event or procedure. CISs are developed with the input of key groups including the patient, clinicians and other relevant stakeholders, to identify the most important information.[19] A CIS differs from a core outcome set (COS), in that a COS is a set of outcomes that are measured or reported, as a minimum, in studies relating the subject. A CIS focuses on information points that should be discussed with patients. Despite these differences, the methodology for producing the respective sets is similar, allowing for adoption of the techniques.[19] Having a CIS to discuss with pregnant women will ensure that they are well informed about the decision they are making. Previous CISs have been used as catalysts for discussion about procedures, to improve content of information leaflets and improve decision aids.[17]

We aim to produce a CIS for VB that can be used to facilitate informed choice by improving discussions and enhancing tools used to help women with decision-making about birth choices. This CIS will be useful for anyone planning or considering having, a VB. This will support preparation for VB, enabling better birth experiences and greater satisfaction, as well as enhancing women's post-natal well-being. Here, we describe the protocol for development of the CIS.

## Aim

To develop a CIS for VB to support informed decision-making for childbirth.

## METHODS/DESIGN

### Design

We will conduct a systematic scoping review, qualitative interviews and use a modified Delphi technique to identify the key information for the CIS. The Delphi process has been used in prior CIS studies[17 18] and involves multiple survey rounds of key stakeholder groups to condense the long list of information points, reaching a consensus on a CIS for VB.

### Methods

This project has been given a favourable ethics opinion by the University of Bristol (UOB) Research Ethics Committee (Ref: 10530).

The study is registered with Core Outcome Measures in Effectiveness (COMET) (https://comet-initiative.org/Studies/Details/2069) and reporting adheres to the COS-STAndards for Development, and COS-STAndardised Protocol Items (COS-STAP) have been used to guide the development of the methods.[20 21] A modified COS-STAP checklist for the VB CIS is included in online supplemental appendix 1. The COS-Standards for Reporting will be used to report the CIS.[22]

While there is no standardised way to develop a CIS,[23] existing CIS have adapted methods from the development of COS.[17 18] The COMET handbook and methods from existing CIS have been used to design this study.[17–19 24]

### Scope of the CIS

The CIS will be applicable to women who are considering SVB. SVB is an unassisted VB, without vacuum or forceps, with a natural onset not through medical intervention.[25 26] We will exclude information points pertaining to the decision-making process of induction of labour, instrumental delivery, CB and VB after caesarean section (VBAC).

## Patient and public involvement

Patient and public involvement (PPI) is key to this study, with involvement of a wide range of stakeholder groups in the development and undertaking of this study. Women's perspectives, and those of their partners, are pivotal to each stage of this study, including the qualitative interviews, Delphi survey and its development, and consensus meeting. The study steering committee contains patient representation, ensuring the development of the project is guided by those that the work will affect.

## Study overview

The development of this CIS will take place in four stages (figure 1).

## Stage 1: identification of key information points

A long list of information points pertaining to VB will be generated through three methods:

a. Scoping review of the literature: Information points about VB from published studies will be extracted as well as from patient information leaflets provided to women in the NHS and other trusted organisations.

b. Interviews with women: qualitative interviews will be conducted with pregnant and postnatal women to determine what information is valued by women.

c. Stakeholder survey: An online survey of stakeholders (women, partners, healthcare professionals, interested groups and medicolegal experts) to identify important information for women.

## Scoping review

### Search

Information about VB will be identified from systematic reviews of pregnancy and childbirth literature and supplemented by a review of patient information leaflets. A pragmatic search strategy will be developed by an information specialist to identify outcomes reported in studies, and key issues women identify, about VB. Studies included in the review will be systematic reviews, both qualitative and quantitative. Search terms will include vaginal birth, vaginal delivery, birth, childbirth, natural childbirth, labour, labour, parturition. Searches will be applied to the following electronic databases: CINAHL, Embase, MEDLINE, Web of Science, DoPHER, Epistemonikos, Health Evidence and the Cochrane Library. Searches will be limited to systematic reviews published in the English language. The date is limited to 2020–2022 for the papers, and to 2017–2022 for the Cochrane reviews, due to the large volume of papers that would meet the

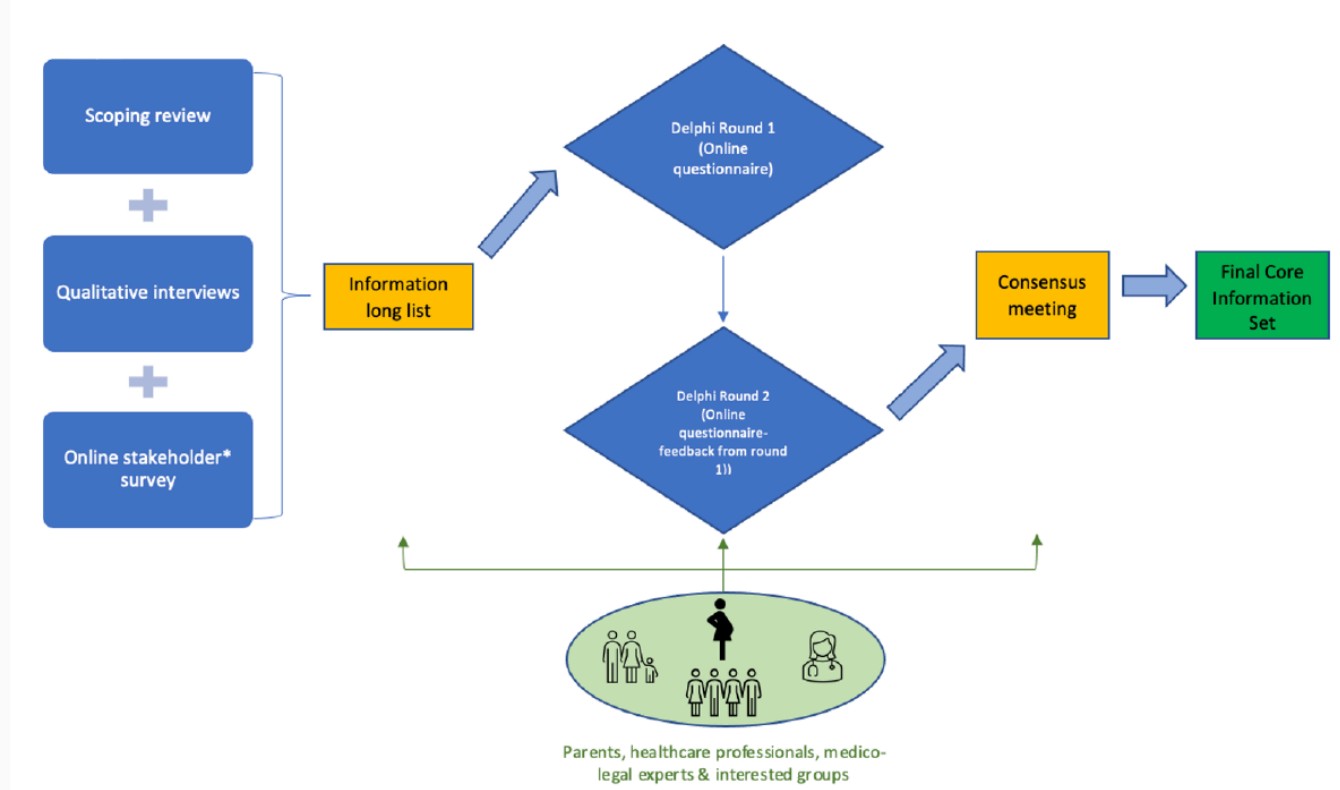

**Figure 1** Study overview diagram. Diagram showing the individual stages of the study and how they feed into the next. Also demonstrates where PPI and stakeholder input will be key in the study. PPI, patient and public involvement.

criteria, thus meaning data saturation from the literature is likely to be achieved within this sample.

We will search for patient information leaflets on VB, to extract information points from these. We will search the online leaflet collections of the Royal College of Obstetricians and Gynaecologists, Tommy's Pregnancy Information and NHS trusts with an online database of leaflets. Leaflets relating to assisted VB (AVB) or VBAC will be excluded from this review, as items relating to these are likely to differ from those for SVB and we hope to develop separate CIS for these in future.

### Paper selection

Screening of titles, abstracts and full texts will be carried out by two members of the research team using systematic reviewing software (Covidence).[27] Any disagreements will be resolved by a third reviewer. We will include all qualitative, quantitative and mixed-method systematic reviews that include information points or outcomes for VB. No studies will be excluded based on judgement of methodological quality or risk of bias assessment as the purpose of the scoping review is to identify information points and outcomes relating to VB, not to examine the quality of data.

### Data extraction

Data will be extracted using a predesigned form, piloted by two members of the research team on a sample of included studies. Data will include basic publication details such as author, publication date, study setting and methodology. A predesigned, and piloted, form will be used for data extraction from leaflets. Extracted data will include format of leaflet, source and year of publication. Outcomes and information points will be extracted verbatim. It will not be necessary to conduct a risk of bias analysis.

### Qualitative interviews
### Method
Semistructured interviews.

### Sample
We will recruit up to 20 women who are >12 weeks pregnant, and/or have had a baby, including those who have experienced an intrauterine death (diagnosed prelabour or during labour), in the last 24 months. Sample size for these interviews will be determined by data saturation being reached, whereby interviews will end when no new themes are generated. Women will be recruited online, via social media and by adverts in community centres. Initial screening questions regarding mode of birth will be devised to allow for purposive sampling of women with a range of birth experiences and preferences including VB, CB and AVB. This will ensure that we gather information points from women from diverse stand points in terms of their decision-making journey to VB. Those who are unable to speak English or consent will be excluded from recruitment due to lack of funds for translation.

### Interviews
Informed consent will be gained using an online form, hosted on REDCap.[28] A topic guide will be developed with the study steering group (online supplemental appendix 2). Interviews will be conducted by a qualitative researcher, recorded on an encrypted audio recorder and uploaded on to the secure UOB server. At the start of the interview, demographic questions will be asked. Participants will receive a £10 voucher to compensate for their time and inconvenience.

### Analysis
Audio recordings of the interviews will be transcribed and analysed by using NVivo software, using thematic analysis.[29] Two members of the research team will read the transcripts initially to familiarise themselves with the process. Interviews will be independently coded by the two members of the research team, to identify important information points about VB from the words of participants. Transcripts will be reread, and each researcher's codes compared for each interview. Codes will then be grouped to create key themes, and key information points will be identified from each of these.[30] Analysis while the interviews are underway will ensure that when data saturation is reached, no additional interviews will be conducted unnecessarily.[31–33]

### Stakeholder survey
### Method
An online qualitative survey will be used to identify further information items by gathering key stakeholder views on what information is important for women to receive about VB.

### Sample
Stakeholders include antenatal and postnatal women, their partners, healthcare professionals who work alongside women in labour and postnatally, representatives from groups with an interest in women's birthing rights and medicolegal experts with an interest in childbirth. We will aim to engage approximately 200 survey participants, recruited online through social media and in community spaces. This sample size has been decided on to ensure that a wide range of views and experiences are captured. We will aim for half of the responses to be from parents whose experiences and views are central to this work.

### Data collection
The survey will be carried out online using REDCap.[28] The survey will include information on the participant's role and demographics, including if they have previously experienced a VB.

To identify what information they believe is vital to provide to women about VB, a free text box will be used. All participants in the survey will be invited to participate in the Delphi process. Participants will be entered into a prize draw to win a voucher worth £10 (5 vouchers available).

## Analysis

Survey data will be thematically analysed by two researchers and key information points extracted. These information points will be added to those from the scoping review and interviews.

## Stage 2: developing the long list of information points and think-aloud interviews

### Developing the long list of information points

Following the identification of key points, we will produce an exhaustive list of information items relating to VB, known as a 'long list', which will feed into the initial stage of the Delphi process.[34 35] Two members of the team will use all information items from the scoping review, interviews, and stakeholder survey to generate the comprehensive list. Information items will be organised into categories for coherence. Examples of categories for information points which may be included are for points relating to maternal physical health, maternal mental health or baby health. Categories may be removed or created dependent on the information points that are discovered.

A meeting will be held to finalise the long list for the Delphi process and establish how the items should be presented and worded in the Delphi survey. This meeting will involve members of the research team. The research team is made up of doctors, midwives, medicolegal experts, a patient representative and researchers. During this meeting, categories will be discussed for each item. Additionally, definitions will be decided to ensure they are understandable for all participants. Where items are similar, group discussions and decisions will take place to ensure there is no duplication.

### Piloting the Delphi survey

Up to 10 think-aloud interviews will be undertaken to test the usability of the Delphi survey and refine the long list of items.[36] The think-aloud method is a well-recognised validation technique for developing an instrument or tool and has been used by COS developers to improve Delphi questionnaires.[19 37] This method is a form of cognitive interviewing that asks participants to verbalise their thoughts as they engage with a task, in this case the survey.[38] We will observe how participants interpret the information items and definitions, and how they interact with the scoring system, identifying any issues. Participants' thoughts are used to improve and perfect the survey to ensure it is understandable and easy to complete. We will record the length of time the survey takes, ensuring it is not too long, therefore, minimising answer fatigue and maximising responses.[19]

After completing a consent form, participants will be asked to trial the Delphi survey using the online platform. Interviews will last 30 min to 1 hour and take place online via teleconferencing software. The interviews will be audio recorded on an encrypted device, and the interviewer will take notes. Overall, 2–3 initial interviews will take place, and comments regarding the survey will be noted, with the audio recordings used if needed to remind about

key points. The comments will then be used to improve the survey, before carrying out the next 2–3 surveys and repeating the process. The approach is repeated until no further changes are indicated, which we estimate will be after approximately 10 interviews.

## Stage 3: the Delphi survey

### Delphi overview

We will aim to conduct two iterative Delphi rounds, followed by a consensus meeting. The survey will be conducted online using REDCap hosted by the UOB.[28]

### Participants and sample size

Participants in the Delphi process will include:

1. Women and their partners:
   Any woman who is over the age of 18, who is currently pregnant, or has had a baby in the last 5 years, and partners of these women.
2. Healthcare professionals
   Those who work with, or provide care for, women, during their pregnancy, during childbirth and/or postnatally (eg, obstetricians and gynaecologists, including trainees, midwives, midwifery support workers, physiotherapists, anaesthetists, general practitioners).
3. Medicolegal experts
   Legal professionals with a specialist interest in reproductive health.
4. Representatives from interested groups/stakeholders
   Representatives from organisations/groups with an interest in women's birthing rights (eg, birthrights, maternity action).

### Sample size

We will obtain a sample with a broad range of experience, an approach supported by the COMET guidelines and previous studies.[17 24] There is no specified size of stakeholder sample for CIS or COS studies.[18 23 24] Some have estimated that the drop-out rate for a Delphi process is around 20%.[35] To account for this potential attrition, we will aim to engage at least 200 participants in the Delphi process. We will aim for at least half of these participants to be antenatal and postnatal women, as their views are vital for this CIS. When gathering responses, if we feel some stakeholder groups are underrepresented, more targeted recruitment for the survey will take place to increase representation.

### Delphi rounds

A two-round modified Delphi process will be conducted online, with each round lasting for 2–8 weeks, dependent on engagement.[17 28] Participants will be asked to complete a consent form before completing the survey, as well as demographic information. Participants will score each survey item for importance for inclusion from 1 to 9 (1–3—limited importance; 4–6—important but not critical; 7–9—critical).[24 39] A priori consensus criteria will be applied.[34] Information items to be carried through to the subsequent round will require ≥80% of participants from either patient/representative group (women, partners,

representatives of interested groups) or the professional group (healthcare workers, medicolegal experts) to score it as critically important (7–9) and <15% from any group to classify it as limited importance. Items will not be carried forward if ≥80% of one group believe it is of limited importance, and <15% believe it is critically important. Where items are rated as critical by ≥80% of participants in one stakeholder group, or there is no consensus, they will be carried forward to the next round.

In round 2, participants who took part in the first survey will be asked to recomplete the survey. They will be informed of their own scores and the median scores of patient/representative and professional groups using both numeric data and graphical illustrations of theirs and others' responses. Items carried through to the consensus meetings will be selected based on the above criteria. If many of the information points meet the criteria to be carried forward to either the second Delphi round or the consensus meeting, the steering group will meet to review the criteria. The steering committee consists of doctors, midwives, researchers, reproductive health lawyer and a patient representative.

In this situation, an adapted form of the OMECRACT framework may be used to organise items.[40 41] Based on this methodology, if many points meet the inclusion criteria, items will be grouped into categories to determine importance and need to be included. Concentric circles indicate the hierarchy of information points (figure 2s). The central ring will be 'absolutely vital information to be included in the CIS'. A further two groups for information points will be used, with a middle ring being 'useful information to have available but may not be vital for inclusion' and an outer ring for items that 'may or may not need to be discussed dependent on the personal circumstances or preferences.' If many items are retained following the application of this criteria, then holding more than one consensus meeting will be considered to ensure adequate time to discuss all points.

We will determine the level of attrition bias by comparing scores of stakeholders who completed both rounds with scores of those who completed only the first round.[19] For transparency, we will record and report the number who have only completed the first round.

### Stage 4: the consensus meeting to decide the final CIS

Following the second Delphi round, an online consensus meeting, using teleconferencing software, will take place with all participants from the second round of the Delphi

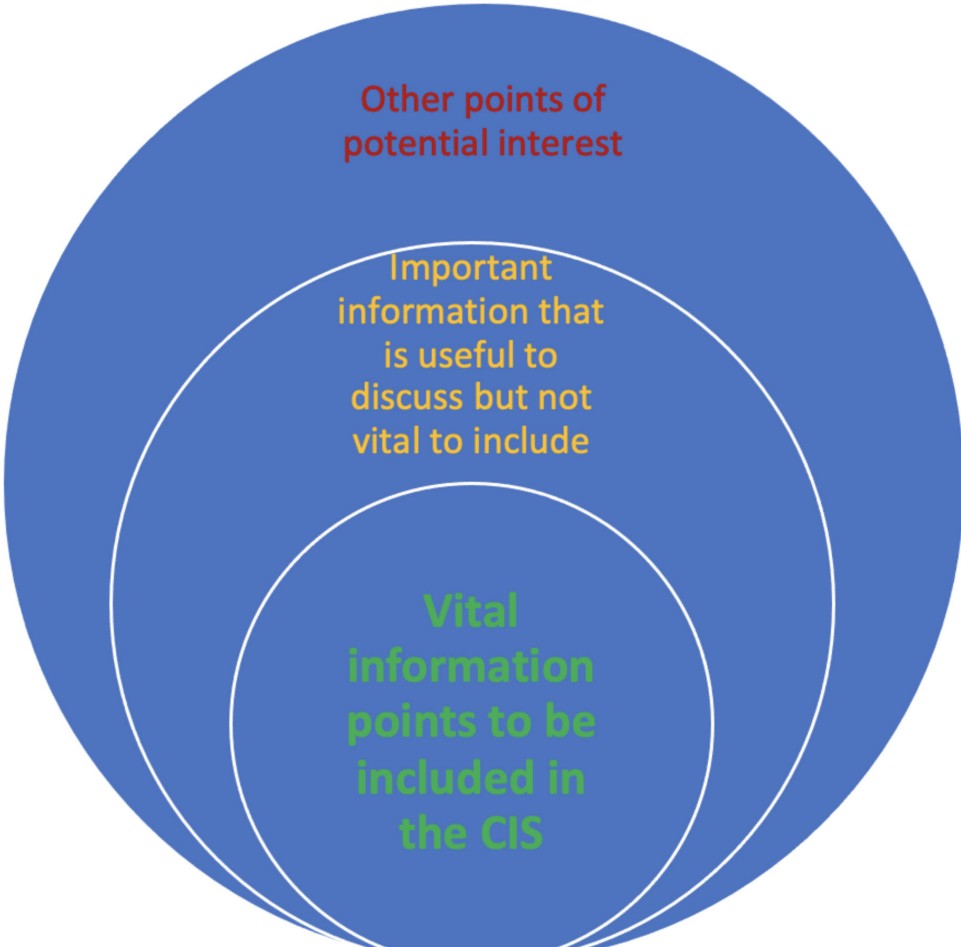

**Figure 2** Omeract onion framework. This diagram demonstrates the framework which framework may be used to organise information items discussed at the consensus meeting. CIS, core information set.

invited. If a large number of the invited stakeholders accept the invitation, we will aim to have the minimum representation from each stakeholder group.[19] Screening questions will allow for purposive sampling to ensure diverse representation of views within these groups. We will aim to recruit 8–10 participants from each stakeholder group, as this number has been sufficient in previous COS.[19 42] A modified nominal technique will be used as a template to drive discussions and reach a group consensus allowing for all participants' views to be heard in the discussion.[43]

During this meeting, items carried through from the second Delphi round will be discussed and descriptive statistics created for each domain and displayed to participants, to stimulate discussion, with median scores for each item presented as graphs.[18 34]

The participants will anonymously rate the retained information domains from the second round using online voting software.[18] Items will be categorised as 'consensus in—information point to be included in final CIS', 'consensus out—item not included in the final CIS' or 'no consensus—items for which opinions on inclusion are divided'. Consensus will be defined as ≥80% of participants agreeing on the vote for an item. Where consensus is not reached, there will be further discussion and additional voting. Scores of those participating in the consensus meeting will be compared with those who only participated in the first two Delphi rounds, to assess if the consensus meeting views are representative of those who only completed the survey.[19] When consensus is reached for all items, we will have the final CIS.

There is no definitive amount of information points that should be included in a CIS. Of the published CIS, the number of information points included in the final CIS have ranged from 8 to 13.[17 18] We will, therefore, aim to have a maximum of 15 information points in our CIS. While it is important the list is concise, we recognise that the unique nature of VB means we may end up with more than 15 points, so there will be flexibility in the acceptable size of the list.

## DISCUSSION

VB is physiological and carries benefits but is not without risk. Informed consent requires that women are well informed when making birth choices. We will develop a CIS using a systematic, transparent, process to help to standardise the information given to women planning to have a VB. It will be informed by what women and other stakeholders believe is the most important information to be discussed, which, if taken up, will ensure all have equitable access to this information.

This will be the first published CIS regarding VB. The study's biggest strength is PPI which is central to the work, both in study development and methodology, meaning the CIS will represent the voices of the women it is aiming to empower. Unfortunately, due to time and financial constraints, interviews to develop the long list will only include women who speak English and those who have given birth in the past 5 years, potentially missing some information items. Other stakeholder groups will also not be interviewed. As this study aims to represent women's voices, interviewing women seems most appropriate. The scoping review and survey should mitigate this limitation, by identifying long-term outcomes and representing other groups.

This CIS can be used as the catalyst for discussions about VB, to promote truly informed birth choices. With this CIS, we will have worked out what information is important. Future work will be needed to elaborate on the information points, adding statistical data to support understanding of risks and benefits, from national databases, systematic literature reviews and expert input. The CIS can also be used to inform patient information leaflets and decision-making tools.

## Ethics and dissemination

The dissemination strategy for the CIS will be developed with the steering committee and Policy Bristol to ensure it is maximally utilised to improve discussions women have with healthcare professionals regarding their birth choices. The findings of the CIS will be published in a peer-reviewed specialty journal. An overview of the CIS will be shared with the COMET initiative, with whom the study is registered. The results will be presented at both national and international conferences of relevant bodies.

For the purpose of open access, the author has applied a Creative Commons 124 Attribution (CC BY) licence to any Author Accepted Manuscript version arising.

Ethical approval has been granted by the University of Bristol Research Ethics Committee (Ref: 10530).

**Acknowledgements** We would like to thank the Research England Policy Support Fund (PSF) for funding this project.

**Contributors** ADemetri and AM conceived the study, obtained the funding drafted the protocol and manuscript. ADavies drafted the protocol and manuscript. DB, CB, AS, DL and EM helped with drafting the manuscript and formulating the methods for the study. SD, SI, SM, CDS, GC, ADempsey and GS assisted with conceiving and drafting the manuscript for stage 1. GB, GC and AJ helped with study conception and feasibility. All authors read and approved the final manuscript.

**Funding** This study was funded by the Research England Policy Support Fund (PSF).

**Disclaimer** The sponsors have played no role in designing the study, data collection, analysis, interpretation of data, writing of the report and decisions for publication.

**Competing interests** None declared.

**Patient and public involvement** Patients and/or the public were involved in the design, or conduct, or reporting, or dissemination plans of this research. Refer to the Methods section for further details.

**Patient consent for publication** Not applicable.

**Provenance and peer review** Not commissioned; externally peer reviewed.

responsibility arising from any reliance placed on the content. Where the content includes any translated material, BMJ does not warrant the accuracy and reliability of the translations (including but not limited to local regulations, clinical guidelines, terminology, drug names and drug dosages), and is not responsible for any error and/or omissions arising from translation and adaptation or otherwise.

**ORCID iDs**
Andrew Demetri http://orcid.org/0000-0002-2820-5919
Anna Davies http://orcid.org/0000-0003-0743-6547
Danya Bakhbakhi http://orcid.org/0000-0003-1906-5069
Sharea Ijaz http://orcid.org/0000-0001-5727-1790
Abi Merriel http://orcid.org/0000-0003-0352-2106

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
