## [Reviewer comments · BMJ Open]

ARTICLE DETAILS

TITLE (PROVISIONAL)	Vaginal Birth Core Information Set: Study protocol for a Delphi study to achieve a consensus on a 'core information set' for vaginal birth
AUTHORS	Demetri, Andrew; Davies, Anna; Bakhbakhi, Danya; Ijaz, Sharea; Dawson, Sarah; McGuinness, Sheelagh; Beasor, Gemma; Clayton, Gemma; Johnson, Abigail; De Souza, Chloe; Dempsey, Aine; Snook, Gabriella; Sharp, Andrew; Lissauer, David; Mcgoldrick, Emma; Burden, Christy; Merriel, Abi

VERSION 1 – REVIEW

REVIEWER	Kingdon, Carol University of Central Lancashire, Research in Childbirth and Health Group
REVIEW RETURNED	14-Dec-2022

GENERAL COMMENTS	Background Paragraph 1 (lines 150-156) introduces the importance of informed choice in maternity care but makes no reference to existing studies of choice and birth method, or decision aids for women offered the choice between a vaginal or a caesarean birth. These are relevant and should inform purposive sampling decisions in the design. Please add. From the UK, they include:  • Farnworth DA, Robson SC, Thomson RG, Burges Watson D, Murtagh MJ. Decision support for women choosing mode of delivery after a previous caesarean section: a developmental study. Patient Educ Couns. 2008;71:116–24. • Frost J, Shaw A, Montgomery A, Murphy D. Women's views on the use of decision aids for decision making about the method of delivery following a previous caesarean section: qualitative interview study. BJOG. 2009;116:896–905 • Kingdon C, Neilson J, Singleton V, Gyte G, Hart A, Gabbay M, Lavender T. Choice and birth method: mixed-method study of caesarean delivery for maternal request. BJOG. 2009 Jun;116(7):886-95. Lines 185-197 introduce previous research investigating core outcome sets^{13,16} and core information sets.^{14,15} This protocol is for the development of a Core Information Set. Appendix 1 is the COS-STAP (Core Outcome Set-STAndardised Protocol Items). While the checklist fields and application of Delphi methods are transferable, it could be argued that there are differences in the research purpose and product of studies. For clarity, please add a sentence to highlight similarities and differences between core outcome sets and core information sets. Methods/Design
--

	Lines 296-298 state we will recruit up to 20 women who are >12 weeks pregnant, and/or women who have had a baby, including those who have experienced an intrauterine death, in the last 24 months. Please clarify in the protocol will women experiencing an intrauterine death have had the loss of their baby confirmed pre-labour or during labour? Lines 301-304 describe how women will self-select for recruitment and will then be purposively sampled. To capture the range of birth experiences and/or preferences distinguish between spontaneous vaginal, instrumental vaginal, elective caesarean and emergency caesarean section. It will be hard to achieve with 20 women but consider more configurations of preferences and experiences evident in the literature. For example, planned vaginal - actual MOB vaginal. Planned caesarean - actual MOB vaginal. No preference - actual MOB vaginal. No preference - actual MOB emergency CS. There is a typo in line 304 (word missing). Line 401 to 404 details how representatives from organisations and groups who have an interest in women's birthing rights, and medico-legal experts will be sought as participants via social media. Will there be purposive sampling of these participants to reflect the range of views evident in current debates about birth method on social media? Similarly, lines 446-448 states we will determine the level of attrition bias by comparing the score of stakeholders who completed both Delphi rounds. For transparency, please commit to record and report. General This is an important study. It would be helpful if the authors could commit to the timelines for completion in the protocol. Thank you.
--	---

REVIEWER	Black, Mairead University of Aberdeen I currently lead an NIHR-funded study (Plan-A) developing a decision aid to support mode of birth planning in routine maternity care. A number of study aspects are similar to those described in this manuscript.
REVIEW RETURNED	05-Jan-2023

GENERAL COMMENTS	Abstract Introduction could be improved by stating that 'A CIS could improve the quality of information given regarding planned mode of birth options..' to make clear that the information provided should inform decision-making but is not proven to do so in this context. Introduction Line 188 – missing a word 'make' Methods Line 224 missing the words 'be' 'the' Line 237 missing the word 'a' Line 266 I wonder if 'operative vaginal birth' and relevant terms are missing Line 274 – it would be helpful to clarify if this will include resources specific to operative vaginal birth, vaginal birth after previous
--

	caesarean or vaginal birth in any other specific group e.g. in maternal diabetes, macrosomia, pre-eclampsia etc. Line 283 is missing the word 'of' Line 304 is missing the word 'to' Line 304. It would be helpful to make clear how you will 'gather information points from diverse stand points' in terms of your planned recruitment process. Will you devise a set of recruitment screening questions based upon set criteria? Line 311 has an unnecessary 'for' in it. Of note the definition of CIS provided in the introduction states that the content demonstrates what is important to 'all relevant stakeholders', but the interview component only includes women. It seems likely that key considerations from healthcare professionals and partners may therefore be omitted. It is noted that the target survey sample size is 200, of whom at least 100 will be parents. It would be helpful to understand how representation of the other 5 stakeholder groups will be determined to make up the other max 100 participants. Given the wide range of health professionals managing the consequences of vaginal birth (from birth to adulthood in the case of children, and from birth to death in the case of women), it will be important to ensure balance in representation of views. Line 360. Who are the 'research team' and does this include public representation? This will be critical to ensure that the list of outcomes presented in the survey reflects what is important to the public. Line 399 – should say 'including' not 'include;' It is notable that general practitioners are not included in the Delphi (or other) study plans. Considering that GPs support management of various consequences of childbirth in both women (physical and psychological) and offspring (e.g. where brain injury or nerve injury has occurred) this would seem to be an important gap. Line 409 – 'the' should read 'there' Given the importance of the steering committee in resolving issues with the Delphi, who makes up the Steering committee – what proportion of the group represent the public? Line 455 – how will you decide who not to include in the consensus meeting? What if there are too many items to discuss in one meeting? How will you manage this? The aim to limit the information points to 15 requires substantial justification. Given the unique nature of vaginal birth in affecting two individuals in different ways, and potentially having life-long effects (both physical and psychological) it seems that an arbitrary maximum of 15 points could undermine the integrity of the study to a large extent. Will you simply remove items of critical importance if there are more than 15? It is unclear throughout the manuscript whether the CIS will specifically target 'planned' vaginal birth rather than actual vaginal birth. It is intimated in places but should be made explicit throughout. As it stands study limitations are not discussed at all. This should be addressed.
--	--

VERSION 1 – AUTHOR RESPONSE

- Comment 1: Paragraph 1 (lines 150-156) introduces the importance of informed choice in maternity care but makes no reference to existing studies of choice and birth method, or decision aids for women offered the choice between a vaginal or a caesarean birth. These are relevant and should inform purposive sampling decisions in the design. Please add. From the UK, they include:
 - o Farnworth DA, Robson SC, Thomson RG, Burges Watson D, Murtagh MJ. Decision support for women choosing mode of delivery after a previous caesarean section: a developmental study. *Patient Educ Couns.* 2008;71:116–24.
 - o Frost J, Shaw A, Montgomery A, Murphy D. Women’s views on the use of decision aids for decision making about the method of delivery following a previous caesarean section: qualitative interview study. *BJOG.* 2009;116:896–905
 - o Kingdon C, Neilson J, Singleton V, Gyte G, Hart A, Gabbay M, Lavender T. Choice and birth method: mixed-method study of caesarean delivery for maternal request. *BJOG.* 2009 Jun;116(7):886-95.

Response: Thank you for this insightful comment. I have amended this section to briefly discuss these studies and decision aids which help women decide between caesarean or vaginal birth. I have also referenced the aforementioned studies.

‘Recent work has looked at improving informed choice regarding birth method, as well as developing decision aids for women choosing between vaginal or CB.2–4 To improve the quality of maternity care, further work must be done to ensure women are provided with the information they require to make a fully informed choice about their planned mode of birth.’ [Page number 6, line 270-274]

- Comment 2: Lines 185-197 introduce previous research investigating core outcome sets^{13,16} and core information sets.^{14,15} This protocol is for the development of a Core Information Set. Appendix 1 is the COS-STAP (Core Outcome Set-STANDARDISED Protocol Items). While the checklist fields and application of Delphi methods are transferable, it could be argued that there are differences in the research purpose and product of studies. For clarity, please add a sentence to highlight similarities and differences between core outcome sets and core information sets.

Response: Thank you, I have added a sentence explaining the main difference between a core information and core outcome set, as well as mentioning the similarity in the methods. ‘A CIS differs from a core outcome set (COS), in that COS development produces a set of outcomes that are measured or reported, as a minimum, in studies relating the subject. A CIS focuses on information points that should be discussed with patients. Despite these differences, the methodology for producing the respective sets is similar, allowing for adoption of the techniques.’¹⁹ [Page number 7, line 313-322]

- Comment 3: Lines 296-298 state we will recruit up to 20 women who are >12 weeks pregnant, and/or women who have had a baby, including those who have experienced an intrauterine death, in the last 24 months. Please clarify in the protocol will women experiencing an intrauterine death have had the loss of their baby confirmed pre-labour or during labour?

Response: Thank you for this comment. I have clarified this based on the suggestion to note that both groups (those who have had an intra-uterine death confirmed pre-labour or during labour) will be included.

‘...including those who have experienced an intrauterine death (diagnosed pre-labour or during labour), in the last 24 months.’ [Page number 12, line 433-434]

- Comment 4: Lines 301-304 describe how women will self-select for recruitment and will then be purposively sampled. To capture the range of birth experiences and/or preferences distinguish between spontaneous vaginal, instrumental vaginal, elective caesarean and emergency caesarean section. It will be hard to achieve with 20 women but consider more configurations of preferences and experiences evident in the literature. For example, planned vaginal - actual MOB vaginal. Planned

caesarean - actual MOB vaginal. No preference - actual MOB vaginal. No preference - actual MOB emergency CS.

Response: Thank you for this observation. When people declare their interest to participate in the interviews, we will ask initial screening questions on the type of delivery they have experienced to help with purposive sampling, in order to capture as wide a range of views as possible. Demographic questions asked at the beginning of the interview will be asked regarding planned birth and actual mode of birth. The topic guide for the interviews will explore this also, looking at birth plans. This has been adjusted in the protocol.

'Initial screening questions regarding mode of birth will be devised to allow for purposive sampling of women with a range of birth experiences and preferences including VB, CB and AVB' [Page number 12, line 437-438]

- Comment 5: Line 401 to 404 details how representatives from organisations and groups who have an interest in women's birthing rights, and medico-legal experts will be sought as participants via social media. Will there be purposive sampling of these participants to reflect the range of views evident in current debates about birth method on social media?

Response: Thank you for this helpful comment. As it is an open survey, it will be difficult to tightly control participants and stakeholder Group sizes. However, we will aim to maximise representation of different views by advertising of the Delphi online via email and through social media, trying to approach specific groups, and emailing these directly. We do recognise, however, that this will not be perfect. We will ask for demographic details at the start of the Delphi to allow for analysis of participant backgrounds, therefore highlighting weaknesses in the study that may affect generalisability.

- Comment 6: Similarly, lines 446-448 states we will determine the level of attrition bias by comparing the score of stakeholders who completed both Delphi rounds. For transparency, please commit to record and report.

Response: This is an important point and I have added a sentence to state that we will record and report.

'For transparency, we will record and report the number who have only completed the first round.' [Page number 19, line 611-613]

- Comment 7: This is an important study. It would be helpful if the authors could commit to the timelines for completion in the protocol. Thank you.

Response: I have added start and proposed end date to the protocol.

'Planned start date 1st June 2022. Planned end date 31st August 2023.' [Page number 3, line 78]

Comments from Reviewer 2

- Comment 1: Introduction could be improved by stating that 'A CIS could improve the quality of information given regarding planned mode of birth options..' to make clear that the information provided should inform decision-making but is not proven to do so in this context.

Response: Thank you for this suggestion. I have reworded the introduction of the abstract to incorporate this and to clarify that this is a theoretical benefit of the proposed CIS.

'This CIS could improve the quality of information given regarding mode of birth options, as women will be given information prioritised by patients and stakeholders regarding vaginal birth, empowering them to make informed decisions about their birth. We aim to describe the protocol for the development of this vaginal birth CIS.' [Page number 3, line 61-65]

- Comment 2: Line 188 – missing a word 'make'

Response: Thank you. This has been corrected.

• Comment 3: Line 224 missing the words 'be' 'the'

Response: Thank you. I have added the missing words to this line.

• Comment 4: Line 237 missing the word 'a'

Response: Thank you. This has now been corrected.

• Comment 5: Line 266 I wonder if 'operative vaginal birth' and relevant terms are missing

Response: Thank you for this comment, and it is definitely something that has been discussed with the rest of the team. We opted not to include 'operative vaginal birth' and other similar terms as we see the information items relating to this to be distinctly different to SVB. Potential future projects will look at developing CIS for 'operative vaginal birth' or 'vaginal birth after caesarean section'.

• Comment 6: Line 274 – it would be helpful to clarify if this will include resources specific to operative vaginal birth, vaginal birth after previous caesarean or vaginal birth in any other specific group e.g. in maternal diabetes, macrosomia, pre-eclampsia etc.

Response: Thank you for this observation and I agree that it needs clarification. It did not include resources regarding operative birth or vaginal birth after caesarean as these were viewed to have a different set of prescribed risks/benefits, differing from spontaneous vaginal birth. Those on specific groups were included in the review. I have added a line to explain the exclusion of the aforementioned leaflets.

'Leaflets relating to assisted vaginal birth (AVB) or VBAC will be excluded from this review, as items relating to these are likely to differ greatly from those for SVB.' [Page number 11, line 409-411]

• Comment 7: Line 283 is missing the word 'of'

Response: Thank you. This has now been corrected.

• Comment 8: Line 304 is missing the word 'to'

Response: Thank you. I have now corrected this.

• Comment 9: Line 304. It would be helpful to make clear how you will 'gather information points from diverse stand points' in terms of your planned recruitment process. Will you devise a set of recruitment screening questions based upon set criteria?

Response: Thank you for this observation. We will ask initial screening questions regarding mode of birth once a participant declares interest, to allow for us to purposively sample, maximising diversity of views. This has been clarified in the text.

'Initial screening questions regarding mode of birth will be devised to allow for purposive sampling of women with a range of birth experiences and preferences including VB, CB and AVB.' [Page number 12, line 437-438]

• Comment 10: Line 311 has an unnecessary 'for' in it.

Response: Thank you for pointing this out. I have edited and corrected this.

• Comment 11: Of note the definition of CIS provided in the introduction states that the content demonstrates what is important to 'all relevant stakeholders', but the interview component only includes women. It seems likely that key considerations from healthcare professionals and partners may therefore be omitted.

Response: Thank you for this observation. I agree that this is a limitation of the study, as we will not interview other stakeholder groups in the initial stage, and this has now been noted as a limitation in the 'Strengths and limitations' sections at the start of the manuscript. However, the stakeholder survey will allow us to capture their views. Furthermore, the scoping review, encompassing academic literature, will, be a louder voice for these other groups that it is for women. As this study aims to have

women's voices at the centre, we felt that we should focus our resources on them for the interviews. The 'strengths and limitations' section has been updated to note this limitation.

'The qualitative interviews to help with the formulation of the long list will only be with antenatal and postnatal women, and not the other stakeholder groups. This means that the key considerations from healthcare professionals and partners may therefore be omitted and missing from the list.' [Page number 5, line 228-231]

- Comment 12: It is noted that the target survey sample size is 200, of whom at least 100 will be parents. It would be helpful to understand how representation of the other 5 stakeholder groups will be determined to make up the other max 100 participants. Given the wide range of health professionals managing the consequences of vaginal birth (from birth to adulthood in the case of children, and from birth to death in the case of women), it will be important to ensure balance in representation of views.

Response: Thank you for this observation, and I agree that it is very important. It will be difficult to definitely ensure the views and responses are balanced between the stakeholder groups. During the survey, we will review responses, in particular the amount by each group, and then aim to increase uptake in underrepresented stakeholder groups by targeting advertising/sharing of the survey to these groups, however, we cannot guarantee that we will obtain an even spread of stakeholders.

'When gathering responses, if we feel some stakeholder groups are underrepresented, more targeted recruitment for the survey will take place to increase representation.' [Page number 17, line 571-573]

- Comment 13: Line 360. Who are the 'research team' and does this include public representation? This will be critical to ensure that the list of outcomes presented in the survey reflects what is important to the public.

Response: Thank you for raising this as an important point to clarify. The research team is made up of obstetric doctors, midwives, a medico-legal expert, a patient representative, and researchers (I have added a sentence stating this). Hopefully this wide range of perspectives, and patient representation, will allow the survey to reflect what is important to the public. 'The research team is made up of doctors, midwives, medico-legal experts, a patient representative, and researchers.' [Page number 15, line 516-517]

- Comment 14: Line 399 – should say 'including' not 'include;'

Response: Thank you for pointing this out. I have corrected this.

- Comment 15: It is notable that general practitioners are not included in the Delphi (or other) study plans. Considering that GPs support management of various consequences of childbirth in both women (physical and psychological) and offspring (e.g. where brain injury or nerve injury has occurred) this would seem to be an important gap.

Response: Thank you for flagging this, GPs are an important group to include. Have amended the list of healthcare professionals to include GPs.

'...e.g. obstetricians and gynaecologists, including trainees, midwives, midwifery support workers, physiotherapists, anaesthetists, general practitioners.' [Page number 16, line 556-558]

- Comment 16: Line 409 – 'the' should read 'there'

Response: Thank you for pointing this out. I have corrected this.

- Comment 17: Given the importance of the steering committee in resolving issues with the Delphi, who makes up the Steering committee – what proportion of the group represent the public?

Response: The steering committee is made up of obstetric doctors, midwives, researchers, and a patient representative.

'The steering committee consists of doctors, midwives, researchers, and a patient representative.' [Page number 18, line 695-697]

• Comment 18: What if there are too many items to discuss in one meeting? How will you manage this?

Response: This is an important point. As outlined on page 18, paragraph 5, line 515-526, the OMERACT onion model will be used if there are many information points to determine those to be carried forward to the consensus meeting. If many of these points are retained still and deemed vital to be included and discussed at the consensus meeting, then we may have to hold more than one meeting. I have added a line to reflect this.

'If many items are retained following the application of this criteria, then holding more than one consensus meeting will be considered to ensure adequate time to discuss all points.' [Page number 18, line 606-608]

• Comment 19: Line 455 – how will you decide who not to include in the consensus meeting?

Response: We will use purposive sampling to decide who will be included if we have lots of interest. We will ask demographic questions and select participants to ensure a wide range of views are represented, including those who have experience different mode of birth and are from different backgrounds.

'Screening questions will allow for purposive sampling to ensure diverse representation of views within these groups.' [Page number 19, line 620-621]

• Comment 20: The aim to limit the information points to 15 requires substantial justification. Given the unique nature of vaginal birth in affecting two individuals in different ways, and potentially having life-long effects (both physical and psychological) it seems that an arbitrary maximum of 15 points could undermine the integrity of the study to a large extent. Will you simply remove items of critical importance if there are more than 15?

Response: Thank you, this is a good point. As stated, previous CIS have had a maximum of 15 points, but it is correct to say the vaginal birth differs from these previous studies and should be treated as such. The aim will be to have 15 points, but if it becomes evident during the process that more may be needed for the CIS, then it will be adapted, either to increase the number of points included, or to merge points for conciseness. A line has been added to reflect this in the text.

'Whilst it is important the list is concise, we recognise that the unique nature of VB means we may end up with more than 15 points, so there will be flexibility in the acceptable size of the list.' [Page number 20, line 647-649]

• Comment 21: It is unclear throughout the manuscript whether the CIS will specifically target 'planned' vaginal birth rather than actual vaginal birth. It is intimated in places but should be made explicit throughout.

Response: Thank you for this comment, it is an important observation. This CIS will be used for those who a planning to have a vaginal birth, but the information points will hopefully be relevant for anyone who experiences a vaginal birth, planned or unplanned. A line has been included in the background to clarify this.

'This CIS will be useful for anyone planning, or considering having, a VB.' [Page number 8, line 329]

• Comment 22: As it stands study limitations are not discussed at all. This should be addressed.

Response: Thank you for this comment. A paragraph has been added to the discussion section of the manuscript, describing the main strengths and limitations of the study.

• This will be the first published CIS regarding VB. The study's biggest strength is PPI which is central to the work, both in study development and methodology, meaning the CIS will represent the voices of the women it is aiming to empower. Unfortunately, due to time and financial constraints, interviews to develop the long list will only include women who speak English and those who have given birth in the past 5 years, potentially missing some information items. Other stakeholder groups will also not be interviewed. As this study aims to represent women's voices, interviewing women seems most

appropriate. The scoping review and survey should mitigate this limitation, by identifying long-term outcomes, and representing other groups.’ [Page number 20-21, line 658-670]

Following these revisions, minor adjustments have also been made to the manuscript to ensure that the abstract and main body of text meet the word limits of the journal.

Thank you once again for all the comments on the manuscript, and your consideration. We look forward to hearing from you in due time regarding our submission and to respond to any further questions and comments you may have.

VERSION 2 – REVIEW

REVIEWER	Kingdon, Carol University of Central Lancashire, Research in Childbirth and Health Group
REVIEW RETURNED	03-Apr-2023
GENERAL COMMENTS	The authors have addressed all my earlier comments. Thank you.

VERSION 2 – AUTHOR RESPONSE